# Optimization of Photothermal Therapy Treatment Effect under Various Laser Irradiation Conditions

**DOI:** 10.3390/ijms23115928

**Published:** 2022-05-25

**Authors:** Donghyuk Kim, Hyunjung Kim

**Affiliations:** Department of Mechanical Engineering, Ajou University, Suwon-si 16499, Korea; kimdonghyuk20@ajou.ac.kr

**Keywords:** apoptosis, bioheat transfer, gold nanoparticles, laser condition, photothermal therapy, squamous cell carcinoma, thermal damage

## Abstract

The photothermal effect refers to a phenomenon in which light energy is converted into heat energy, and in the medical field, therapeutics based on this phenomenon are used for anticancer treatment. A new treatment technique called photothermal therapy kills tumor tissue through a temperature increase and has the advantages of no bleeding and fast recovery. In this study, the results of photothermal therapy for squamous cell carcinoma in the skin layer were analyzed numerically for different laser profiles, intensities, and radii and various concentrations of gold nanoparticles (AuNPs). According to the heat-transfer theory, the temperature distribution in the tissue was calculated for the conditions under which photothermal therapy was performed, and the therapeutic effect was quantitatively confirmed through three apoptotic variables. In addition, the laser intensity and the volume fraction of AuNPs were optimized, and the results provide useful criteria for optimizing the treatment effects in photothermal therapy.

## 1. Introduction

The photothermal effect refers to the phenomenon in which light energy is converted into thermal energy [1,2]. This effect is utilized in various applications, e.g., for measuring thermal properties [3,4,5] and for laser hair removal, scar removal, and anticancer treatment in the medical field [6,7,8]. In photothermal therapy, tumor tissue is killed by increasing its temperature using a laser (light energy) in the presence of a photothermal conversion agent. One advantage of this method is that there is no bleeding, and the patient recovers faster compared with conventional treatment techniques based on incision [9,10].

Because photothermal therapy causes tumor death by increasing the temperature, it is necessary to understand the death mechanism of the biological tissue with respect to the temperature. Depending on the temperature, the mechanism of cell tissue death is either necrosis or apoptosis [11,12]. Necrosis occurs at ≥50 °C, and cell death occurs through the leakage of intracellular contents to the outside. In the case of a tumor, if it dies because of necrosis, the intracellular contents leak into the surrounding tissues, leading to a risk of cancer metastasis. Conversely, apoptosis refers to a form of self-destruction without affecting surrounding tissues and occurs between 43 and 50 °C. If the tumor tissue undergoes necrosis, as mentioned previously, there is a risk of cancer metastasis. Therefore, in photothermal therapy, it is important to kill the tissue through apoptosis by maintaining an appropriate temperature range.

In photothermal therapy, biological tissues absorb different levels of laser energy according to the wavelength bands [13,14,15]. For visible-light lasers, biological tissue absorbs large amounts of energy owing to its large light-absorption coefficient; therefore, it exhibits a large temperature increase. In addition, the temperature of the surrounding tissue excessively increases through conduction heat transfer, which causes the tissue temperature to increase beyond the temperature band corresponding to apoptosis to the temperature band where necrosis occurs. In contrast, while using an infrared laser, the light-absorption coefficient of the biological tissue is small; accordingly, the increase in the temperature of the tissue is small, owing to the minimal absorption of light energy. However, because this leads to a situation where the temperature range corresponding to apoptosis cannot be reached, photothermal agents are administered to the tumor tissue for increasing the temperature to the desired range. The photothermal agents are materials that increase the light-absorption coefficient at a specific wavelength and have the advantage of being able to increase the absorption rate for a desired laser wavelength. They are composed of various materials, and among them, gold nanoparticles (AuNPs) are widely used for reasons such as their harmlessness to the human body [16,17].

Considering these factors, many researchers investigate photothermal therapy. Fan et al. [18] confirmed the effect of photothermal therapy on subcutaneous cervical cancer in mice using indocyanine green (IR820) as a photothermal and fluorescence agent. To improve the photothermal effect and biocompatibility of IR820, the amphiphilic polymer Pluronic F-127 was used to form IR820@F-127 nanoparticles through self-assembly. The IR820@F-127 nanoparticles exhibited a high photothermal conversion efficiency of 35.2% and excellent photothermal stability as well as excellent biocompatibility, cell absorption ability, and bioimaging ability for tumor sites. The study was conducted in vitro and in vivo, and the therapeutic effect on cervical cancer was confirmed using a 793 nm laser. Chen et al. [19] developed plasmonic nanoparticle clusters (PNCs) to fabricate highly efficient photothermal agents. Au-based PNCs using FeOOH were used, and the resonance frequency and absorption efficiency were calculated by combining the effective medium approximation theory and full-wave electrodynamic simulations. In in vitro and in vivo experiments targeting human breast cancer cells (MCF-7), mouse breast cancer cells (4T1), and human cervical cancer cells (HeLa), a photothermal conversion efficiency of up to 84% was achieved under the optimized cluster growth condition. Li et al. [20] developed new nanoparticles to increase the efficiency of photothermal treatment and address the issue of cancer recurrence owing to the lack of photothermal nanomaterials with high photothermal conversion efficiencies. Nanoparticles (Ag@Ag_2_O/LDHs-U) in which Ag@Ag_2_O core–shell and ultrathin layered double hydroxide (LDH) structures were combined to improve the photothermal performance significantly in the NIR-II region were developed. The results of high-resolution transmission electron microscopy, extended X-ray absorption fine structure spectroscopy, and X-ray photoelectron spectroscopy confirmed that the ultrafine Ag@Ag_2_O core–shell nanoparticles were highly dispersed and firmly fixed on the ultrathin LDH nanosheet. The particles exhibited a photothermal conversion efficiency of 76.9% at a wavelength of 1064 nm, and in vitro and in vivo experiments confirmed that Ag@Ag_2_O/LDHs-U had excellent biocompatibility and phototherapy efficacy in the NIR-II region. Ren et al. [21] conducted a numerical analysis about irradiating a laser onto biological tissue in photothermal therapy using gold nanoparticles. Through the Monte Carlo method and Beer’s law, the heat generation of biological tissue to which gold nanoparticles were injected was calculated, and the temperature distribution inside the tissue was calculated. Wang et al. [22] created a numerical analysis model for the tumor injected with GNPs and confirmed the temperature distribution of the tumor located on the skin surface and surrounding normal tissues through the Monte Carlo method through numerical analysis. Maksimova et al. [23] confirmed the effect of photothermal therapy by confirming the temperature distribution and the absorbed laser distribution when 808 nm and 810 nm lasers were irradiated to animals treated with a silica/gold nanoshell and malignant tumors through experiments and numerical analysis. This study confirmed the destruction of tumor cells according to the intensity and type of various lasers, and through this, it was confirmed that photothermal therapy can effectively destroy tumor cells.

Recent research on photothermal therapy has mainly been directed toward developing new photothermal agents to overcome the low photothermal conversion efficiency of existing photothermal agents. Although photothermal therapy utilizes the photothermal effect, which is based on a heat-transfer theory, related theoretical studies are insufficient. In particular, lasers do not have a uniform light intensity in the radial direction owing to their optical characteristics and have various light intensity distributions. However, studies for simultaneously confirming the photothermal therapy effect according to the various profiles, intensities, and radii of the laser and concentration of photothermal agents are insufficient. Additionally, in photothermal therapy, the treatment effect was simply confirmed phenomenologically, or the degree of apoptosis in tumors, and the amount of thermal damage to surrounding normal tissues was not quantified. Therefore, in the present study, the amounts of laser heat absorbed and scattered in the medium when various lasers were applied to the skin layer containing squamous cell carcinoma were investigated through the Monte Carlo method, and the temperature distribution in the medium was calculated through a numerical analysis using the thermal-diffusion equation. In addition, using the apoptotic variable proposed by Kim et al. [24], the extent to which the temperature range corresponding to apoptosis in the tumor was maintained and the amount of thermal damage to the surrounding normal tissue were quantified to determine the conditions that produce the optimal treatment effect under various laser irradiation conditions.

## 2. Results and Discussion

### 2.1. Temperature Distribution of Tissue for Different Laser Profiles

Figure 1 shows the temperature of the tumor center with respect to the treatment time when the *f_v_* of injected AuNPs was 10^−6^, and the laser intensity was 100 mW. Figure 1a,b presents the results for the Gaussian and top-hat profiles, respectively. When the laser radius ratio ($\varphi_{r}$) was 1, the radii of the tumor and laser were equal, and when $\varphi_{r}$ was 0.5, the radius of the laser was 50% of that of the tumor. As shown in the graph, the temperature at the tumor center varied with respect to $\varphi_{r}$. In general, the temperature of the tumor center was higher when $\varphi_{r}$ was lower. This is because the tumor tissue absorbs more energy because of the increase in the energy per unit area as $\varphi_{r}$ decreases. In addition, the temperature was lower for the top-hat profile than for the Gaussian profile because the energy density of the center was higher in the former case. Therefore, the temperature distribution of the tumor and surrounding normal tissues was confirmed for different laser profiles, laser radii and intensities, and *f_v_* values of injected AuNPs. Furthermore, the maintenance degree of the apoptosis temperature range during the treatment period as well as the amount of thermal damage to surrounding normal tissues was quantitatively analyzed to suggest treatment conditions.

### 2.2. Apoptosis Retention Ratio

In this study, the apoptosis retention ratio ($\theta_{A}^{*}$) was used to determine the degree of apoptosis in tumor tissue quantitatively [24]. $\theta_{A}^{*}$ is obtained by calculating the ratio of the total tumor volume to the volume corresponding to the apoptosis temperature range of 43–50 °C for each total treatment time and then calculating the average of the values. Through this variable, the degree of maintenance at 43–50 °C (a temperature range known to induce apoptosis in the given treatment period) was quantitatively confirmed. In this study, the time used for the calculation of $\theta_{A}^{*}$ was selected as 600 s, which is the total treatment time.

Figure 2 shows the results for $\theta_{A}^{*}$ with respect to the $\varphi_{r}$ for each *f_v_* of AuNPs when a laser with a Gaussian profile was used. The intensity of the laser (*P_l_*) at which $\theta_{A}^{*}$ was maximized increased with $\varphi_{r}$ for all *f_v_* values. This is because the apoptosis temperature band can only be induced by applying a high *P_l_* because the energy absorbed per unit area decreases as $\varphi_{r}$ increases. In addition, the maximum value of $\theta_{A}^{*}$ decreased as $\varphi_{r}$ decreased. This is because as $\varphi_{r}$ decreased, the temperature increased excessively and exceeded the apoptosis temperature range owing to intensive heating in a narrow area of the tumor tissue. Additionally, the maximum value of $\theta_{A}^{*}$ increased as *f_v_* decreased. This is because as *f_v_* decreased, the depth to which the laser penetrated the medium increased, and heating occurred in a wide area in the depth direction, and excessive energy absorption did not occur because the absorption coefficient was relatively small. For the laser with a Gaussian profile, the apoptosis temperature band was maintained for the longest time under the conditions of $\varphi_{r}$ = 1.75, *f_v_* = 10^−6^, and *P_l_* = 210 mW.

Figure 3 shows the results for $\theta_{A}^{*}$ with respect to $\varphi_{r}$ for each *f_v_* of AuNPs when a laser with a top-hat profile was used. The overall trend is similar to that for the Gaussian profile, and the maximum value of $\theta_{A}^{*}$ was larger. This is because the tumor tissue was uniformly heated along the radial direction, as energy was uniformly absorbed along each laser radius. For the laser with a top-hat profile, the apoptosis temperature band was maintained for the longest time under the conditions of $\varphi_{r}$ = 1.25, *f_v_* = 10^−6^, and *P_l_* = 170 mW. In contrast to the case of the Gaussian profile, the optimal point for $\theta_{A}^{*}$ depended on the $\varphi_{r}$. This is because the amount of energy absorbed per unit area decreased as the $\varphi_{r}$ increased, and after the optimal radius ratio was reached, the laser energy was insufficient for achieving the apoptosis temperature range.

### 2.3. Thermal Hazard Retention Value

When a NIR laser is applied to tumor tissue for treatment, the laser energy is absorbed by the tumor tissue owing to the increased light-absorption coefficient resulting from the AuNP injection. Subsequently, the absorbed heat is transferred to the surrounding normal tissue through conduction, increasing the temperature of the surrounding normal tissue. To perform selective treatment of the tumor tissue, the thermal damage to the surrounding normal tissue must be minimized. Therefore, in this study, the amount of thermal damage to normal tissue near the tumor was quantitatively analyzed using the thermal hazard retention value ($\theta_{H}^{*}$) [24]. $\theta_{H}^{*}$ uses weights given to phenomena expressed in each temperature band in biological tissue as shown in Table 1. In addition, as in Equation (1), the ratio of the weighted sum of the volumes corresponding to each temperature band to the total volume of the surrounding normal tissue at each time step is calculated. After that, the calculated values as in Equation (2) are taken as the average of the total treatment time.



(1)
$$\theta_{H}=\frac{\sum_{j=1}^{m}\left(V_{n}(T)\cdot w_{j}\right)}{V_{n}}$$


(2)
$$\theta_{H}^{*}=\frac{1}{\tau }\int_{0}^{\tau }\theta_{H}(\tau )d\tau$$



$\theta_{H}^{*}$ quantitatively identifies the average amount of thermal damage during the treatment as the ratio of the volume of surrounding normal tissues to the volume of thermal damage after weighting various phenomena occurring in biological tissues according to the temperature.

Figure 4 shows $\theta_{H}^{*}$ with respect to $\varphi_{r}$ for each *f_v_* of AuNPs and each laser profile. In the case of the Gaussian profile, the thermal damage was maximized when $\varphi_{r}$ was 0.75. For $\varphi_{r}$ ≤ 0.75, the amount of thermal damage to normal tissue around the tumor tissue was relatively small because the laser energy was focused on the center of the tumor tissue. Therefore, $\theta_{H}^{*}$ was smaller than that for $\varphi_{r}$ = 0.75. For $\varphi_{r}$ ≥ 0.75, the heat transfer amount to the surrounding normal tissues was reduced because the applied laser energy per unit area decreased. In the case of the top-hat profile, similar trends were observed, and the thermal damage was maximized when $\varphi_{r}$ was 1. Additionally, $\theta_{H}^{*}$ decreased as *f_v_* decreased. This is because the medium absorbed a smaller amount of heat owing to the reduced light-absorption coefficient as *f_v_* decreased.

For both laser profiles, $\theta_{H}^{*}$ was minimized when $\varphi_{r}$ was 1.75, that is, when the radius of the laser significantly exceeded that of the tumor tissue. However, from a treatment perspective, $\theta_{A}^{*}$ should also be considered simultaneously. For Gaussian lasers, the thermal damage to the surrounding normal tissues is minimized when $\varphi_{r}$ is 1.75, and $\theta_{A}^{*}$ is maximized when $\varphi_{r}$ is 1.75, even when considering the degree of apoptosis in tumor tissues. However, for the top-hat laser, the thermal damage to the surrounding normal tissues is minimized when $\varphi_{r}$ is 1.75, but $\theta_{A}^{*}$ is maximized when $\varphi_{r}$ is 1.25. However, $\theta_{H}^{*}$ increases with the laser intensity, and $\theta_{A}^{*}$ has the intensity of the laser showing the maximum value for each $\varphi_{r}$. Accordingly, because the intensities for both laser profiles must be considered simultaneously, the optimal treatment conditions must be identified by simultaneously considering $\theta_{A}^{*}$, $\theta_{H}^{*}$, and the laser intensity.

### 2.4. Effective Apoptosis Retention Ratio

Previously, $\theta_{A}^{*}$ and $\theta_{H}^{*}$ were used to confirm quantitatively the maintenance of the apoptosis temperature in tumor tissues and the amount of thermal damage to surrounding normal tissues. However, because apoptosis in the tumor tissue and thermal damage to the surrounding normal tissues occur simultaneously during actual treatment, the results were confirmed through the effective apoptosis retention ratio ($\theta_{eff}^{*}$) [24]. $\theta_{eff}^{*}$ was calculated as the ratio of $\theta_{A}^{*}$ to $\theta_{H}^{*}$, and using this parameter, the optimal treatment conditions were confirmed for different laser conditions and *f_v_* values of injected AuNPs.

Figure 5 shows $\theta_{eff}^{*}$ with different $\varphi_{r}$ values for each *f_v_* of AuNPs and each laser profile. Simultaneously confirming the degree of apoptosis in tumor tissues and the thermal damage amount of surrounding normal tissues revealed that $\theta_{eff}^{*}$ was maximized when $\varphi_{r}$ was 1 for both laser profiles. This indicates that $\theta_{A}^{*}$ was maximized when $\varphi_{r}$ was 1.75 using the Gaussian profile and 1.25 using the top-hat profile, respectively, under the two laser profile conditions, but high thermal damage to the surrounding normal tissues occurred owing to the high *P_l_* at the optimal point. For the Gaussian profile, the optimal treatment conditions were as follows: $\varphi_{r}$ = 1, *f_v_* = 10^−6^, and *P_l_* = 130 mW. For the top-hat profile, they were $\varphi_{r}$ = 1, *f_v_* = 10^−5^, and *P_l_* = 120 mW.

From a heat-transfer viewpoint, setting $\varphi_{r}$ to 1 for both laser profiles was identified as the optimal treatment condition. However, if minimizing thermal damage is prioritized over the therapeutic effect from a clinical perspective, a condition in which the amount of thermal damage is smaller may be preferential even if less therapeutic effects are obtained. In cases where the amount of thermal damage is minimized, treatment may be impossible. Therefore, clinically, information on the reference point for the amount of thermal damage and required treatment time is necessary, and the results of this study are useful for establishing such reference points.

## 3. Materials and Methods

### 3.1. Monte Carlo Simulation and Heat-Transfer Model

In Monte Carlo simulation, laser particles absorbed and scattered in a medium, e.g., biological tissue, can be considered, and the behavior of the laser particles can be estimated stochastically using random numbers [27]. The Monte Carlo simulation was performed by calculating the moving angle and distance of a particle. First, the angle of movement of the particles can be determined using the deflection angle in Equation (3) and the azimuth angle in Equation (4), where *ξ*, *θ*, *g*, and *ψ* represent selected random numbers, deflection angles, anisotropy coefficients, and azimuth angles, respectively.
(3)$$\cos\theta =\left\{\begin{array}{cc} \frac{1}{2g}\left\{1\;+\;g^{2}\;-\;{\left[\frac{1\;-\;g^{2}}{1\;-\;g\;+\;2g\xi }\right]}^{2}\right\} & if\;g>0\\ 2\xi -1 & if\;g=0 \end{array}\right.$$
(4)ψ=2πξ

After the deflection angle and azimuth angle are calculated, they are converted into a vector representing the direction in which the particle moves in the Cartesian coordinate system through Equations (5)–(7), where $\mu_{x}$,$\;\mu_{y}$, and $\mu_{z}$ represent the direction cosines of the axes.
(5)$$\mu_{x}^{\prime }=\frac{\sin\theta }{\sqrt{1-\mu_{z}^{2}}}(\mu_{x}\mu_{z}\cos\psi -\mu_{y}\sin\psi )+\mu_{x}\cos\theta$$
(6)$$\mu_{y}^{\prime }=\frac{\sin\theta }{\sqrt{1-\mu_{z}^{2}}}(\mu_{y}\mu_{z}\cos\psi -\mu_{x}\sin\psi )+\mu_{z}\cos\theta$$
(7)$$\mu_{z}^{\prime }=-\sin\theta \cos\psi \sqrt{1-\mu_{z}^{2}}+\mu_{z}\cos\theta$$

If the calculated angle is perpendicular to the tissue surface, it can be converted into Equations (8)–(10) along each axis.
(8)$$\mu_{x}^{\prime }=\sin\theta \cos\psi \;(0\;<\;\theta \;<\;\pi ,\;0\;<\;\psi \;<\;2\pi )$$
(9)$$\mu_{y}^{\prime }=\sin\theta \sin\psi \;(0\;<\;\theta \;<\;\pi ,\;0\;<\;\psi \;<\;2\pi )$$
(10)$$\mu_{z}^{\prime }=\left\{\begin{array}{cc} \cos\theta & if\;\mu_{z}>0\\ -\cos\theta & f\;\mu_{z}<0 \end{array}\right.\;(0\;<\;\theta \;<\;\pi )$$

The moving distance of the particle is calculated using Equation (11) using the total attenuation coefficient of the medium and the selected random number. The total attenuation coefficient of the medium is the sum of the absorption coefficient and scattering coefficient of the medium, as indicated by Equation (12), where *S*, $\mu_{tot}$, $\mu_{a}$, and $\mu_{s}$ represent the distance traveled by photons once, total attenuation coefficient, light-absorption coefficient, and light-scattering coefficient, respectively.
(11)$$S=\frac{-\ln(\xi )}{\mu_{t}{}_{ot}}$$
(12)$$\mu_{t}{}_{ot}=\mu_{a}+\mu_{s}$$
(13)$$\Delta W=W\frac{\mu_{a}}{\mu_{t}{}_{ot}}$$

Finally, when the angle and distance of movement are calculated, the ratio of energy attenuation when the particle moves once is calculated using Equation (13). Here, *W* is the energy weight of the particle, and according to the attenuation ratio based on the absorption of the laser particle, the particle moves until the energy of the particle converges to zero. The light-absorption distribution of the laser in the medium can be calculated by repeatedly calculating the number of specified particles in this process.

The temperature distribution inside the medium was calculated using the thermal-diffusion equation, i.e., Equation (14), utilizing the light-absorption distribution calculated via the Monte Carlo simulation and the thermal properties of the medium. Here, *k*, *ρ*, and $c_{v}$ represent the thermal conductivity, density, and specific heat, respectively, and τ, *F*, and $P_{l}$ represent time, the fluence rate, and the laser intensity, respectively [28].
(14)$$\begin{array}{c} \Delta T=\frac{\Delta \tau }{\rho c_{v}}\left(\mu_{a}FP_{l}dxdydz+(T_{x^{-}}-T)\frac{2kk_{x^{-}}}{k+k_{x^{-}}}\frac{dydz}{dx}\right.+(T_{x^{+}}-T)\frac{2kk_{x^{+}}}{k+k_{x^{+}}}\frac{dydz}{dx}\\ +(T_{y^{-}}-T)\frac{2kk_{y^{-}}}{k+k_{y^{-}}}\frac{dxdz}{dy}+(T_{y^{+}}-T)\frac{2kk_{y^{+}}}{k+k_{y^{+}}}\frac{dxdz}{dy}\\ \left.+(T_{z^{-}}-T)\frac{2kk_{z^{-}}}{k+k_{z^{-}}}\frac{dxdy}{dz}+(T_{z+}-T)\frac{2kk_{z+}}{k+k_{z^{+}}}\frac{dxdy}{dz}\right) \end{array}$$

### 3.2. Optical Properties of AuNPs and Medium

Biological tissues exhibit a small light-absorption coefficient for lasers in the NIR region. This results in the poor absorption of heat from the laser which becomes an obstacle to reaching the temperature range required for tumor death. To compensate for this, in photothermal therapy, a photothermal agent that increases the light-absorption coefficient at a specific wavelength is injected into tumor tissue to enhance the laser heat absorption. In this study, AuNPs, which increase the light-absorption coefficient at a specific wavelength via localized surface plasmon resonance, were used as photothermal agents [29].

The optical properties of AuNPs can be calculated using Equation (15) [30], where *f_v_*, *Q*, and *r_eff_* represent the volume fraction of injected AuNPs, optical efficiency, and light-absorption area, respectively. The optical efficiency of the AuNPs was calculated using the discrete dipole approximation [31,32]. In this study, rod-type AuNPs with an effective light-absorption area of 20 nm and an aspect ratio of 6.67 were used. After the optical properties of AuNPs are calculated, the optical properties of the biological tissue injected with AuNPs can be calculated using Equation (16) [33].
(15)$$\mu_{a,np}=0.75f_{v}\frac{Q_{a}\;}{r_{eff}},\;\mu_{s,np}=0.75f_{v}\frac{Q_{s}\;}{r_{eff}}$$
(16)$$\mu_{a}=\mu_{a,m}+\mu_{a,np},\;\mu_{s}=\mu_{s,m}+\mu_{s,np}$$

### 3.3. Numerical Investigation

First, verification of numerical analysis modeling was performed. The validation of the numerical analysis model was performed through two methods: comparison of results with previous studies and an experiment using a biomimetic phantom. Comparison with previous researchers was compared with the results of Crochet et al. [34]. This study assumed that a spherical tumor with a radius of 5 mm existed in the center of a tissue with a radius of 30 mm and a length of 30 mm as shown in Figure 6a. The radius of the irradiated laser was set to 5 mm and 15 mm, and the intensity was fixed to 1.7 W. The laser irradiation time was fixed at 800 s, and the results of comparison with previous studies are shown in Figure 6b. As shown in the figure, as a result of comparison with previous studies, it was confirmed that the results were consistent within a maximum error of 2%. The verification of numerical analysis modeling through experiments was confirmed in the author’s previous study through a biomimetic phantom [24]. The temperature results were compared between the experiment and numerical analysis, and the average root-mean-square error was 0.1677; therefore, the simulation accurately reflected the actual condition.

In this study, numerical analysis modeling was conducted for a condition under which squamous cell carcinoma occurred inside the skin layer, which was composed of the epidermis, papillary dermis, reticular dermis, and subcutaneous fat. It was assumed that squamous cell carcinoma with a radius of 5 mm and depth of 2 mm was located at 0.1 mm below the skin surface inside normal tissue with a radius of 15 mm and depth of 20 mm, as shown in Figure 7. A 1064 nm wavelength laser was used as the heat source, and the thermal and optical properties of each skin layer and tumor are presented in Table 2.

In the numerical analysis, the profile and radius of the laser and the *f_v_* of injected AuNPs were changed to identify the optimal treatment conditions for photothermal therapy under various conditions. The lasers had Gaussian and top-hat profiles, as shown in Figure 8. The typical laser profile has a Gaussian shape, and in the case of the top-hat profile, an additional device is installed in front of the laser beam to distribute the energy density uniformly according to the beam radius.

As described previously, the top-hat laser applies a uniform energy according to the laser radius. However, because the amount of energy absorbed per unit area varies according to the beam radius, for a given energy level, an appropriate laser radius must be selected. In the case of the Gaussian laser, the energy absorbed by the tumor tissue varies according to the radial direction because of the nonuniformity of the energy density along the radial direction, as shown in Figure 9. If the diameter of the laser is significantly larger than that of the tumor tissue, uniform heating is induced in the entire tumor tissue because the profile of the laser is gradually flattened. However, because the surrounding normal tissue also absorbs laser heat, an appropriate laser radius must be selected to maintain the temperature in the range where apoptosis occurs. In this study, the laser beam diameter was selected based on 1/e^2^ width.

Accordingly, in this study, the degree of apoptosis temperature band maintenance in tumor tissue was confirmed by varying the radius ratio of the Gaussian and top-hat lasers to the tumor. In addition, the therapeutic effect of photothermal therapy was quantitatively confirmed by changing the *f_v_* of injected AuNPs and the intensity of the laser. The treatment time was fixed at 600 s, and the numerical-analysis parameters are presented in Table 3.

The laser profile was set to the Gaussian and top-hat distributions, and the ratio of the laser radius to the tumor radius ($\varphi_{r}$) was varied from 0.5 to 1.75 at intervals of 0.25. The laser intensity was varied in the range of 0–500 mW at 10-mW intervals, and the *f_v_* of injected AuNPs was varied from 10^−3^ to 10^−6^ at intervals of 10^−1^. When AuNPs are injected into tumor tissue, the optical properties of the entire tissue change according to the *f_v_*. In this study, rod-type AuNPs with an effective radius of 20 nm and an aspect ratio of 6.67 were used, and the optical properties of the tumor tissue containing AuNPs for each *f_v_* are presented in Table 4.

## 4. Conclusions

A heat-transfer-based numerical study was performed on photothermal therapy using AuNPs on a skin layer in which squamous cell carcinoma occurred. The behavior of the laser was analyzed using a Monte Carlo simulation, and the temperature distribution in the medium was calculated using the thermal-diffusion equation. Using the calculated temperature distribution, the treatment conditions for photothermal therapy were optimized through $\theta_{A}^{*}$ that quantitatively reflects the degree of apoptosis in tumor tissue, $\theta_{H}^{*}$ that quantifies the thermal damage to surrounding normal tissues, and $\theta_{eff}^{*}$ that simultaneously considers the above two situations.

A numerical analysis was performed by changing the profile, radius, and intensity of the laser and the *f_v_* of injected AuNPs. When a laser with a Gaussian profile was used, the optimal treatment effect was obtained when the $\varphi_{r}$ was 1; the *f_v_* of injected AuNPs was 10^−6^; and laser intensity was 130 mW. For a top-hat laser, the optimal treatment effect was obtained when the $\varphi_{r}$ was 1; the *f_v_* of injected AuNPs was 10^−5^; and laser intensity was 120 mW. Therefore, it can be used as a criterion for improving the treatment effects in photothermal therapy under various laser conditions. In the future, the feasibility of clinical photothermal therapy must be confirmed by verifying the numerical analysis results through in vitro and in vivo experiments. The results of this study can be used as criteria for clinically selecting the degree of thermal damage and treatment effect.

## Figures and Tables

**Figure 1 ijms-23-05928-f001:**
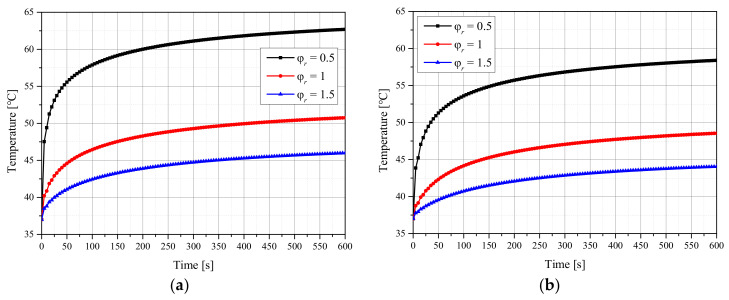
(**a**) Gaussian profile. (**b**) Top-hat profile. Temperature change of the tumor tissue for different radius ratios (*f_v_* = 10^−6^, *P_l_* = 100 mW).

**Figure 2 ijms-23-05928-f002:**
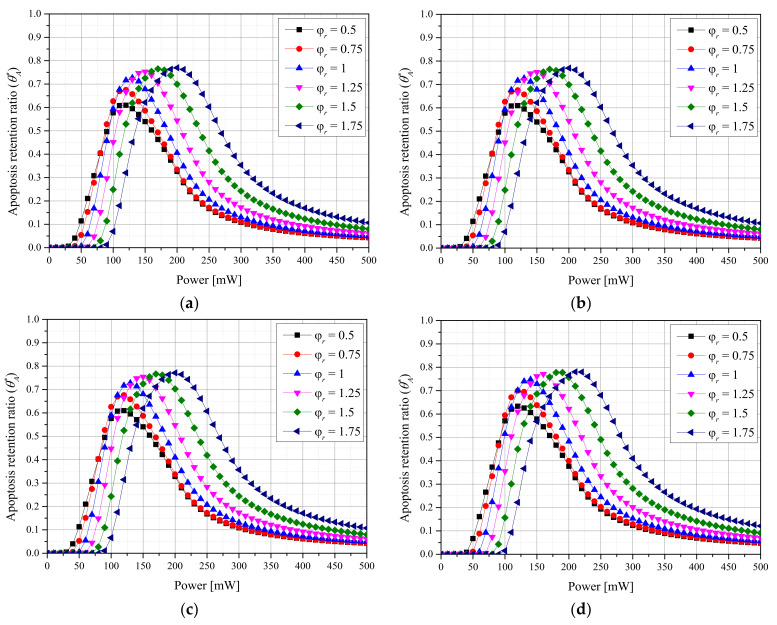
(**a**) *f_v_* = 10^−3^. (**b**) *f_v_* = 10^−4^. (**c**) *f_v_* = 10^−5^. (**d**) *f_v_* = 10^−6^. Apoptosis retention ratio ($\theta_{A}^{*}$) for different radius ratios ($\varphi_{r}$) (Gaussian profile).

**Figure 3 ijms-23-05928-f003:**
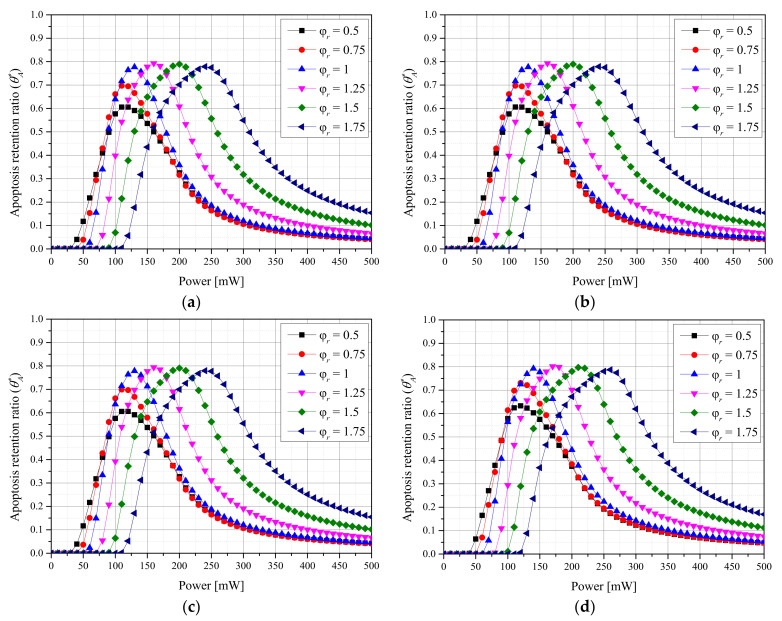
(**a**) *f_v_* = 10^−3^. (**b**) *f_v_* = 10^−4^. (**c**) *f_v_* = 10^−5^. (**d**) *f_v_* = 10^−6^. Apoptosis retention ratio ($\theta_{A}^{*}$) for different radius ratios ($\varphi_{r}$) (top-hat profile).

**Figure 4 ijms-23-05928-f004:**
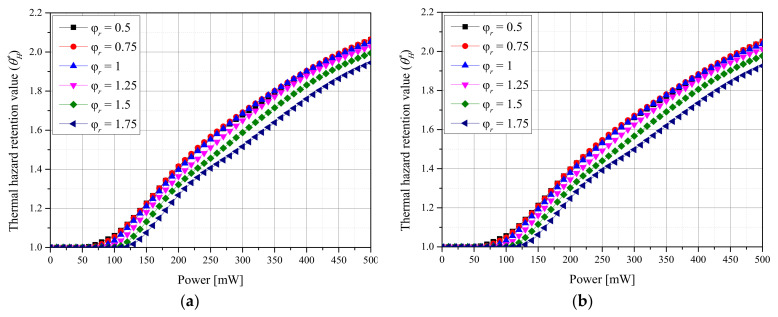
(**a**) Gaussian profile, *f_v_* = 10^−3^. (**b**) Gaussian profile, *f_v_* = 10^−6^. (**c**) Top-hat profile, *f_v_* = 10^−3^. (**d**) Top-hat profile, *f_v_* = 10^−6^. Thermal hazard retention value ($\theta_{H}^{*}$) for different radius ratios ($\varphi_{r}$).

**Figure 5 ijms-23-05928-f005:**
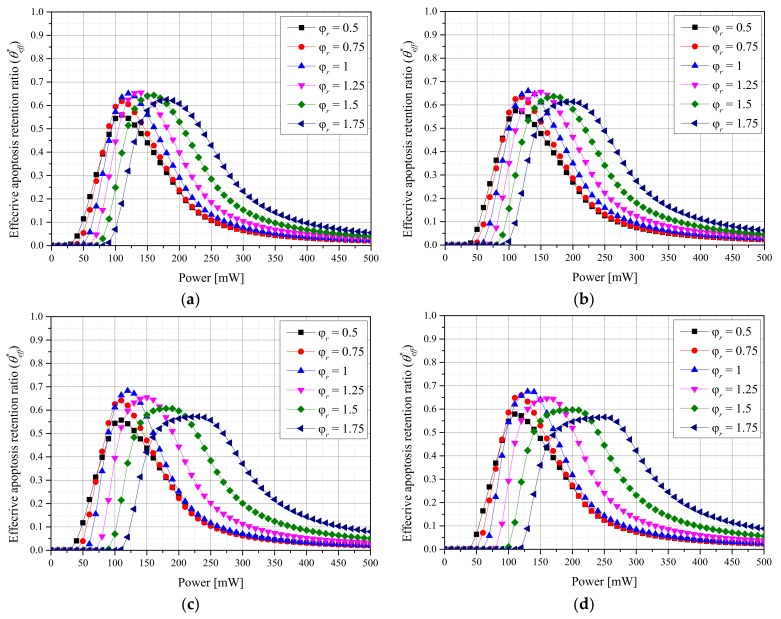
(**a**) Gaussian profile, *f_v_* = 10^−3^. (**b**) Gaussian profile, *f_v_* = 10^−6^. (**c**) Top-hat profile, *f_v_* = 10^−3^. (**d**) Top-hat profile, *f_v_* = 10^−6^. Effective apoptosis retention ratio ($\theta_{eff}^{*}$) for different radius ratios ($\varphi_{r}$).

**Figure 6 ijms-23-05928-f006:**
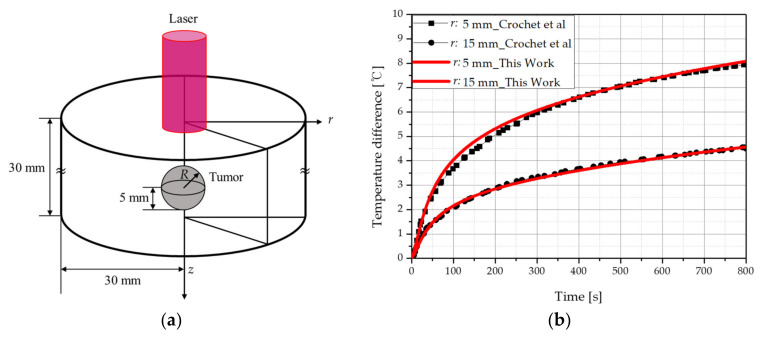
(**a**) Schematic of validation numerical simulation. (**b**) Numerical validation result. Validation of numerical analysis (with previous studies) [34].

**Figure 7 ijms-23-05928-f007:**
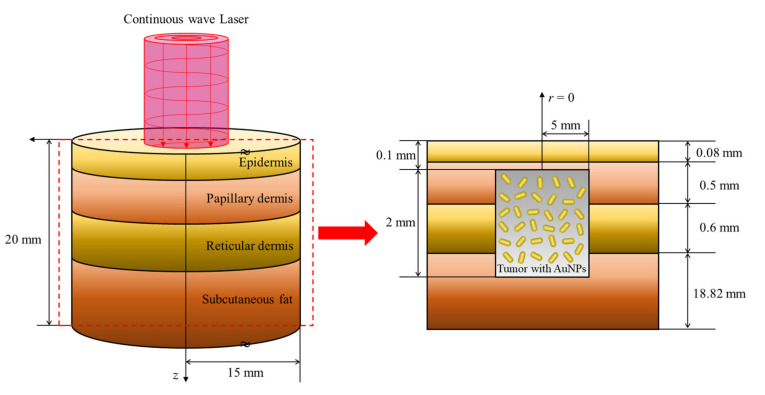
Schematic of the numerical simulation.

**Figure 8 ijms-23-05928-f008:**
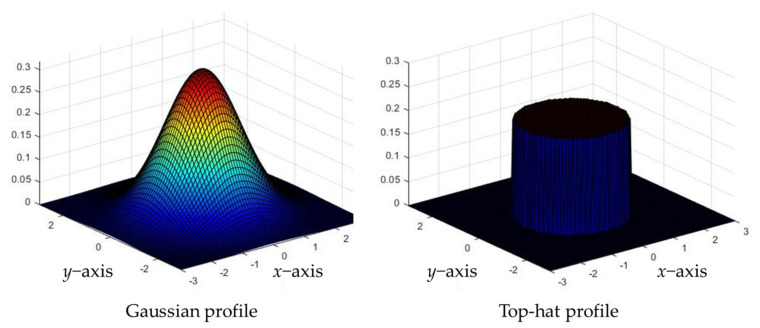
Schematic of laser profiles.

**Figure 9 ijms-23-05928-f009:**
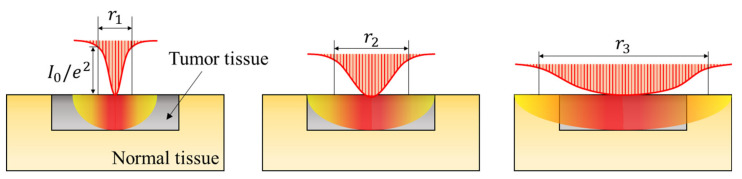
Profile changes for a radius increase in the Gaussian laser profile.

**Table 1 ijms-23-05928-t001:** Laser-induced thermal effects [25,26].

Temperature Range (°C)	Biological Effect	Weight, w
37	Normal	1
37<T<43	Biostimulation	1
43≤T<45	Hyperthermia	2
45≤T<50	Reduction in enzyme activity	2
50≤T<70	Protein denaturation (coagulation)	3
70≤T<80	Welding	4
80≤T<100	Permeabilization of cell membranes	5
100≤T<150	Vaporization	6
150≤T<300	Carbonization	7
T>300	Rapid cutting and ablation	8

**Table 2 ijms-23-05928-t002:** Properties of the skin layers and tumor [35,36,37,38,39,40,41,42].

	t (mm)	k (W/mK)	$c_{v}$ (J/kgK)	ρ (kg/m^3^)	$\mu_{a}$ (mm^−1^)	$\mu_{s}$ (mm^−1^)	g
Epidermis	0.08	0.235	3589	1200	0.4	45	0.8
Papillary dermis	0.5	0.445	3300	1200	0.38	30	0.9
Reticular dermis	0.6	0.445	3300	1200	0.48	25	0.8
Subcutaneous fat	18.82	0.19	2500	1000	0.43	5	0.75
Tumor	2	0.495	3421	1070	0.047	0.883	0.8

**Table 3 ijms-23-05928-t003:** Parameters of the numerical analysis.

Parameter	Case	Number	Remarks
Laser profile type	Gaussian, top-hat	2	
Radius ratio of laser ($\varphi_{r}$)	0.5–1.75	6	Intv: 0.25
Laser power (*P_l_*)	0–500 mW	51	Intv: 10 mW
Volume fraction of AuNPs (*f_v_*)	10^−3^–10^−6^	4	Intv: 10^−1^

**Table 4 ijms-23-05928-t004:** Optical properties of the tumor tissue with different volume fractions of AuNPs.

Volume Fraction of AuNPs	10^−3^	10^−4^	10^−5^	10^−6^
Absorption coefficient ($\mu_{a}$) (mm^−1^)	557.414	55.784	5.62	0.604
Scattering coefficient ($\mu_{s}$) (mm^−1^)	118.575	12.652	2.06	1.001

## Data Availability

Not applicable.

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
