# Peer review of "Optimization of Photothermal Therapy Treatment Effect under Various Laser Irradiation Conditions"

_ijms, 2022, doi:10.3390/ijms23115928_

Round 1
Reviewer 1 Report
In this study, the photothermal therapy for squamous cell carcinoma in the skin layer were analyzed numerically for different laser profiles, intensities, and radii and various concentrations of gold nanoparticles. The detailed comments are as follows:
- What is the definition of diameter of laser with Gaussian profile in the present work? Theoretically, the laser profile has no limits according to the definition of Gaussian profile. Therefore, the laser diameter is usually defined as full width at half maximum or the 1/e2 width or the knife-edge width. Fig. 3 should be revised because it’s a little misleading.
- In Fig. 5, what is the corresponding time for the calculation of apoptosis retention ratio? Is it after the thermal equilibrium is achieved or a specific time, such as 500 s?
- It is not reasonable to use temperature as an indicator for the thermal damage since it also strongly dependent on the time period that the tissue suffers from hyperthermia.
- In the introduction, previous theoretical works about the optimization of temperature distribution or optimization during thermal therapy should be introduced and also to emphasize the novelty of this work.
Reviewer 2 Report
The authors should cite more recent researches and comparing their own data with literature. After addressing this concern this reviewer will consider the paper again.
Reviewer 3 Report
Thanks very much to the editor for giving me this opportunity to review this article. This is a very meaningful article. The idea in this manuscript is novel. This is a carefully done study and the findings are of considerable interest. My detailed comments are as follows: There are many photothermal Therapy Treatments in tumor treatment. Please fully compare this study with previous studies in discussion.
Round 2
Reviewer 1 Report
The authors have addressed all my concerns.
Reviewer 2 Report
Accepted